# Species-specific retention vs. recovery of coral thermal tolerance following nursery propagation
Sebastian Szereday [1,2] ✉, Kok Lynn Chew [1] & Christian R. Voolstra [2] ✉

Thermal screening of coral source material is likely crucial to enhancing long-term restoration success under ocean warming. It is unclear, however, whether reef-based donor colonies retain their thermal tolerance in a nursery environment. Here, we used CBASS acute thermal assays to compare standardized thermal tolerance thresholds (ED50s) of donor colonies from *Acropora cytherea* and *Acropora florida* from two sites in Pulau Lang Tengah, Malaysia to their 'nursery propagules' reared in a common garden coral nursery over 365 days. CBASS assays of reef-based donors and their nursery counterparts were conducted in parallel and over two seasons to assess retention of thermal tolerance following nursery rearing. After 6 months, average ED50s of *A. cytherea* nursery corals were significantly lower compared to their reef-based donor colonies, but such difference disappeared after 365 days. By comparison, no such differences were measurable for *A. florida* and thermal tolerances were retained. Further, we did not observe trade-offs between growth and thermal thresholds for either species. Based on our findings, in situ thermal tolerance differences are likely adaptive and, consequently, either retained or recovered in longer-term restoration settings. Our findings further imply that thermal screening should be conducted prior to nursery propagation to avoid selection based on long-term acclimation artifacts.

Hard corals are the foundational architects and engines of tropical coral reef ecosystems and associated marine biodiversity[1]. Given their high ecological and economic value[2], reef managers, scientists, and stakeholders have been developing and implementing active measures, such as in situ coral restoration[3], to counteract past, present, and anticipated future losses of reef-building hard corals[4–6]. In situ coral restoration predominantly aims at increasing hard coral cover through the propagation of selected coral species[7,8], a practice that has been expanding in scope and popularity, and which is now widely implemented across many of the world's tropical reef regions[9]. However, concurrently with advances in coral restoration practice, ocean warming is resulting in longer, more widespread, and more frequent marine heatwaves[10,11], resulting in severe mass coral bleaching and mortality events in all ocean basins[12,13]. Therefore, long-term coral restoration success is thought to increasingly depend on the preferential selection (screening) of heat tolerant corals[14–16].

Thermal tolerance of hard corals greatly varies across and within species[17–20], geographic reef regions[21,22], reef environments[23,24], and within single reefs[25–27]. Differences in how individual coral colonies respond to heat stress have been associated with their algal symbiont assemblage[28,29],

acclimatory responses following prior exposure to heat stress[30], genetic predisposition[31,32], as well as bacterial associations of the coral host[33,34]. Despite the complex underpinnings of coral thermal tolerance, recent global advances have been achieved to develop standardized diagnostic tests to identify coral phenotypes with superior heat stress tolerance[25,35–37]. However, given that susceptibility to heat stress varies across time and space[38,39], it remains largely unresolved whether heat tolerant donor colonies give rise to heat tolerant nursery corals or whether an acclimatization period to novel coral nursery and reef environments is required. Knowledge regarding this is critical to understand whether coral thermal tolerance is retained in various coral restoration settings, such as coral nurseries[40,41] and following propagation to replenish denuded reef sites[42].

Propagating naturally occurring heat resilient corals to build nurseries and inform restoration is a promising strategy[43]. Preferential selection based on experimentally derived standardized thermal thresholds has been shown to produce multi-species coral nurseries with enhanced ability to survive coral bleaching events[43]. In contrast, without pre-screening, the accidental selection of less resilient corals may impair the heat tolerance of nursery corals[41]. However, the current lack of repeated and parallel thermal stress

[1]Non-profit enterprise for coral reef research and restoration, Coralku Solutions, Kuala Lumpur, Malaysia. [2]Department of Biology, University of Konstanz, Konstanz, Germany. ✉e-mail: sebastian@coralku.org; christian.voolstra@uni-konstanz.de

testing over time of both, reef-based donors and nursery corals is limiting our ability to resolve retention or increase vs. loss and/or subsequent recovery of thermal tolerance in restoration practices[41]. Such testing would also provide clues as to whether the observed thermal tolerance variation is due to adaptive or acclimatory differences.

Although preferential selection of heat tolerant corals for restoration is considered crucial under warming oceans[14,15], this may come with associated risks. For example, compromised growth and reduced survivorship of outplanted thermally tolerant corals have been noted[44,45]. Such trade-offs may stem from associations with thermally tolerant algal symbiont types[46]. However, contrasting evidence exists as well. Co-benefits (i.e., positive associations) between physiological performance traits such as fecundity, growth, and thermal tolerance were found in corals from the Indo-Pacific[47], while cross-tolerance to multiple stressors has been identified for the common Indo-Pacific coral *Acropora millepora*[48]. Thus, trade-offs associated with thermal tolerance may not be universally present across species or universally manifest across environments[24].

To support the optimization of coral restoration outcomes through thermal phenotype pre-screening, we conducted parallel and repeated heat stress testing of nursery and corresponding reef-based donor corals of two hard coral species from two environmentally disparate sites using the Coral Bleaching Automated Stress System (CBASS)[25,35,37]. Parallel testing over time was performed to resolve potential temporal patterns of coral thermal tolerance loss, retention, recovery, or increase. Moreover, putative trade-offs between coral thermal tolerance and coral growth[44,49] during the initial growth-focused nursery phase were investigated.

## Methods

### Study location and coral sampling
We conducted a year-long investigation to assess coral thermal tolerance dynamics at an ongoing restoration site around Pulau Lang Tengah (5°47′ 43.2″N, 102°53′39.7″E), in northeastern Peninsular Malaysia (Fig. 1). In March 2022, reef-based donor colonies of two fast-growing species used within local restoration efforts, *Acropora florida* and *Acropora cytherea* (Henry et al.[50]), were tagged, mapped, and photographed. A total of 40 colonies of *A. florida* and 42 coral colonies of *A. cytherea* from two distinctive reef environments ($n = 20$ and $n = 21$ colonies per site, respectively) were tagged, two replicate fragments from each colony were removed with gardening shears, placed in numbered zip-lock bags, and immediately transferred to a common garden in situ nursery site in a bucket filled with local seawater. Selected donor colonies were visually healthy, showed no signs of disease, bleaching, or partial mortality, and were sampled along a broad depth range between 4.4 and 15.2 m (±1 m tidal range) in a proportionate fashion (i.e., an equal number of colonies per depth zone). A minimum distance of 5 m between adjacent colonies was maintained to maximize phenotypic diversity and to avoid sampling of clonal genotypes[51].

To assess the putative impact of the site of origin on coral thermal tolerance, coral source material was collected from two sites: one where the common garden nursery was established (Site 1) and a second site exhibiting an environmentally disparate reef setting (Site 2) (Fig. 1). Key habitat differences between these reef sites are differential exposure to wind forcing (*sensu* Szereday et al., 2024[27], Site 2 more exposed), difference in diel temperature profiles (Szereday et al., 2024[27], Site 1 exhibits greater diel

temperature variability), and greater micro-habitat heterogeneity at Site 2 due to higher hard coral cover, density, and diversity compared to Site 1[52].

### Common garden nursery
Four coral rebar frames were used as a coral nursery substrate[53] (Fig. 1). The collected fragments from each donor colony were randomly dispersed across the rebar frames and secured with cable ties. To avoid overgrowth and shading within the nursery by adjacent corals, tabular *A. cytherea* were exclusively placed on the bottom half of the rebar frames. Site 1 was selected as the common garden site (Fig. 1) to allow comparison with previous studies that measured growth and survival of corals reared in adjacent rebar and coral tree nurseries[50,54]. The average depth of rebar frames was 9.0 m (i.e., middle point of the frame, top = 8.8 m, bottom 9.2 m) and was kept at the same depth level as the average depth of reef-based donor colonies (i.e., 9.2 m). Each coral fragment was tagged with a rubber tag to trace nursery fragment identity back to the respective reef-based donor colony. Coral nurseries were occasionally cleaned to remove algae and other biofouling organisms during the first 176 days of the study and once again after the northeast monsoon on day 360.

### Coral growth monitoring
Following nursery stocking, coral fragment length, width, and height were measured along the rebar axis (Electronic Supplementary Material, Supplementary Fig. 1) to calculate initial fragment size, expressed as the geometric mean radius (GMR; cm, *sensu* Loya, 1976[55]). The mean initial fragment size for *A. florida* was 3.39 ± 0.13 cm (SEM) and 2.77 ± 0.09 cm (SEM) for *A. cytherea*. Fragment size was re-measured on day 176 (i.e., before the CBASS assays), and the geometric mean radius was used to calculate the specific growth rate (SGR). Specific growth rate is calculated as[50]:

$$SGR = ([\ln(G1) - \ln(G2)] \times 100)/t$$

where t is the total time of growth (i.e., 176 days), with G1 and G2 being the GMR of the final (day 176) and initial size (day 0) in centimeters, respectively.

### Standardized acute heat stress assays using the CBASS platform
To test whether coral thermal tolerance is retained, lost, or increased in coral nurseries, we used the previously established Coral Bleaching Automated Stress Systems (CBASS)[25,35,37] for parallel heat stress testing of nursery corals and their corresponding reef-based donor colonies. Briefly, the CBASS platform consists of four replicate 10 L flow-through tanks with independent temperature profiles customized to prevailing local reef conditions. Here, we selected temperature profiles based on available NOAA climatology (Coral Reef Watch, version 3.1., https://coralreefwatch.noaa.gov/) binned to the nearest 5 km satellite pixel[56]. The control baseline temperature profile was set to 30.5 °C, based on the interpolation of NOAA's historical maximum monthly mean (MMM) temperature of 29.94 °C with local in situ temperature data[27]. The three heat-hold treatment temperatures were then set to MMM + 4 °C (34.5 °C), MMM + 6 °C (36.5 °C), and MMM + 9 °C (39.5 °C) to induce mild, strong, and severe heat stress,

**Fig. 1 | Map of study site and coral nursery location. a** Location of Pulau Lang Tengah (5°47′ 43.2″N, 102°53′39.7″E) along the northeastern coast of Peninsular Malaysia. The location pin in the zoomed-out figure shows the location of the nearest major city (Kuala Lumpur). Site 1 and Site 2 are reef locations from which coral source material was collected, with Site 1 being the location of the common garden nursery. Examples of donor and nursery coral species: (**b**, **c**) *Acropora florida* and (**d**, **e**) *Acropora cytherea*.

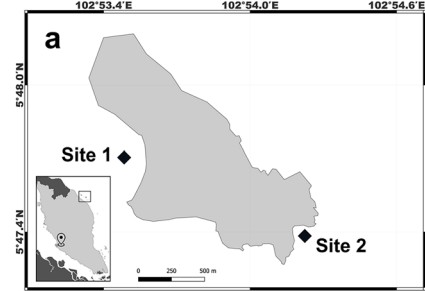
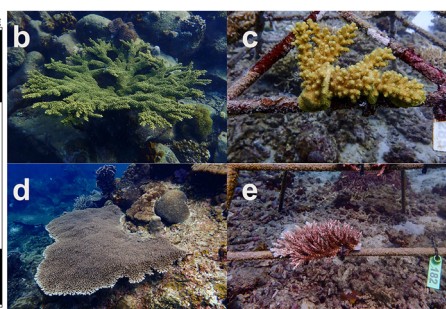

respectively. Heat-hold temperatures were chosen to cover the spectrum of physiological responses to heat stress (from mild to extreme) for subsequent log-logistic modelling of photosynthetic efficiency loss over increasing temperatures, whereby the highest temperature is intended to induce severe bleaching and mortality[36]. Of note, we did not use replicated fragments (tanks) for each temperature treatment, since a prior study highlighted the absence of significant tank effects[25] and to minimize biomass harvesting. Further, we assayed a high number of coral colonies (n = 24 for *A. florida* and n = 27 for *A. cytherea*) to capture biological variation, which also encompasses aspects of technical variance inherent to the experimental workflow.

After coral growth and survival monitoring on day 176 in October 2022, a subset of the total number of colonies was selected to achieve the maximum sample size of n = 14 per CBASS assay. Hereby, 24 nursery corals of *A. florida* (n = 12 from both sites) and 27 nursery corals of *A. cytherea* (n = 14 from Site 1, n = 13 from Site 2) were selected for acute heat stress testing together with their reef-based donor counterparts, representing 24 unique *A. florida* and 27 *A. cytherea* reef-based donor colonies. A total of 192 *A. florida* coral nubbins (i.e., 24 reef-based donors x 4 fragments per colony + 24 nursery corals x 4 fragments per colony) and 216 *A. cytherea* nubbins (27 reef-based donors x 4 fragments per colony + 27 nursery corals x 4 fragments per colony) were collected. The selection of the colony subsets was conducted haphazardly and included fragments of varying color intensity (i.e., from fully pigmented to slightly pale) of variable growth rates (i.e., 0.12–0.34% growth/day for *A. florida*, and 0.07–0.49% growth/day for *A. cytherea*). Of note, selected nursery corals for CBASS assays were on average larger than non-selected corals to ensure sufficient coral biomass for subsequent CBASS assays in March 2023. In further steps, the corresponding reef-based donor colonies were relocated, sampled, and acute heat stress tested to assess standardized thermal tolerances (i.e., ED50s) for both the nursery fragments and donor colonies in parallel (total n = 48 for *A. florida* and n = 54 for *A. cytherea*). Four replicate fragments of approximately 6.0–9.0 cm$^2$ size[57] were collected from each nursery coral and donor colony and randomly distributed across the four temperature treatment tanks, while their position within each tank was further randomized using a random number generator to determine each fragments position. Following the placement of all fragments so that each nursery and in situ coral was in each of the four temperature profiles, a standardized local diel temperature cycle, as per Voolstra et al. (2020)[34], was applied to each experimental tank (Supplementary Fig. 2). The 7 h thermal cycles started at 1 pm and consisted of a 3 h heating phase to the desired heat stress temperature, a 3 h heat-hold phase of the maximum treatment temperature, and a 1 h cooling phase back to the baseline temperature. Temperatures in each treatment tank were controlled with low-cost temperature controllers (InkBird ITC-310T-B) that regulated thermo-electric chillers (IceProbe, NovaTec) and 200 W titanium heaters (Schego). Light was supplied with dimmable 165 W full spectrum LED aquarium lights, set at a 50–50 ratio of blue and white light, respectively, to match in situ reef conditions of approximately 35,000 lux intensity. Before each experiment, HOBO MX2202 Pendant MX temperature and light data loggers (Onset Computer Corporation, USA) were inter-calibrated and then used to continuously monitor temperature and light every minute in each tank throughout the experiment. Local seawater was collected on the morning of each experiment and adjusted to baseline temperature (i.e., 30.5 °C) in a 200 L insulated reservoir tank before the start of the CBASS assays. Once the water temperature was at baseline temperature, all treatment tanks were completely filled, and seawater was continuously supplied to treatment tanks with a submersible aquarium pump at a rate of 25-30 mL/min to achieve 100% turnover of treatment tank volume every six hours. During the 1-hour cooling phase (starting daily at dusk at 6 pm), the white LEDs were first switched off, then the blue LEDs (30 min later), so that all fragments were dark-acclimated. Dark-acclimated photosynthetic efficiency of PSII ($F_v/F_m$) of the coral algae was measured by pulse amplitude modulated (PAM) fluorometry (Diving-PAM underwater fluorometer; Heinz Walz GmbH, Effeltrich, Germany) to derive a physiological measure reflective of heat stress tolerance of corals[58].

In March 2023, 365 days after the first nursery stocking, standardized heat stress assays were again conducted to assess the consistency of coral thermal tolerance. For this, previously tested nursery corals and reef-based donor colonies were resampled for acute heat stress testing, and new fragments from each previously tested nursery and reef-based donor were collected for subsequent CBASS assays. Minor sample size differences (*A. florida* n = 46 in March 2023 vs. n = 48 in October 2022; *A. cytherea* n = 47 in March 2023 vs. n = 54 in October 2022) due to detachment from the nursery, loss of coral tags, and some mortality of reef-based donors during the northeast monsoon are to be noted (Supplementary Table 1). The seasonal timing of acute heat stress testing was deliberate to account for the potential influences of seasonal temperature patterns. Locally, the warmest six-month period of the year is between April and September, and the coolest period occurs between November and February during the annual northeast monsoon, which increases rainfall, cloud cover, wind, and waves, resulting in lower temperatures. In situ temperatures at 1 and 15 m depths were recorded at the common garden nursery site with an Aqualink Smart buoy (Sofar Ocean Technologies Inc., USA). Additionally, in situ temperature data from Site 2 recorded at 8 m depth was partially available for the study period (Supplementary Fig. 3).

## Standardized thermal tolerance threshold assessment – effective dose 50 (ED50)

Coral thermal tolerance thresholds (i.e., Effective Dose 50 - ED50s) are defined as the mean temperature at which 50% of the value measured (here $F_v/F_m$) at baseline temperature is retained[36]. For quantitative comparison of species-specific standardized thermal tolerance thresholds between nursery corals and reef-based donors, photochemical yields ($F_v/F_m$) measurements (see above) were used to fit log-logistic dose-response curves with the DRC package in R (Ritz et al., 2015) using the three-parameter model (LL.3) with model limits for the three parameters 'hill', 'max', and 'ed50' (upper limits = 100, 0.8, NA; lower limits = 10, 0.3, 30) to obtain effective dose 50 (ED50) values[36,59]. For ED50 determination, the measured mean temperatures of the 3 h heat-hold phase were used in lieu of the programmed target temperature (e.g., 31 °C instead of 30.5 °C for baseline, 35 °C instead of 34.5 °C, 37 °C instead of 36 °C, and 40 °C instead of 39.5 °C; measured temperatures of individual heat stress assays are provided in Supplementary Table 2–3). To calculate relative thermal tolerance thresholds, we subtracted the MMM from the determined ED50s to obtain relative thermal thresholds (ED50 – local MMM). This was done separately for each site and condition, i.e., Site 1 - Donor, Site 2 - Donor, Site 1 - Nursery, and Site 2 - Nursery.

## Reproducibility of colony-specific ED50 values and rank/cohort assignments

To support the selection of individual coral colonies for restoration, the reproducibility of thermal tolerance thresholds (ED50s) by means of assignment into bottom and top performing colonies was investigated. The reproducibility of ED50 values on the colony level (i.e., for each genet[60]) was tested by comparing October 2022 ED50s to March 2023 ED50 values for each colony. Additionally, we assessed consistency of cohort assignment (top- vs. bottom-performing colonies) as a function of screening effort (i.e., number of colonies). Colonies were ranked by their ED50s and the mean overlap of the top 5 and bottom 5 colonies comparing donor and nursery corals in October 2022 and March 2023 were determined using random subsampling to assess accuracy as a function of screening effort (number of colonies assayed).

## Statistics and Reproducibility

A linear mixed effect model (LME) was built to measure the contribution of individual response variables with 'temperature' (i.e., in situ measurements 31 °C, 35 °C, 37 °C, 40 °C) and 'condition' (i.e., nursery and donor) set as fixed factors as well as 'site' (i.e., Site 1, Site 2) and 'time' (i.e., October 2022 and March 2023) set as random factors. Since response to heat stress is inherently variable across species[17], each species was tested independently using a separate model iteration. Model analysis was conducted in R Studio

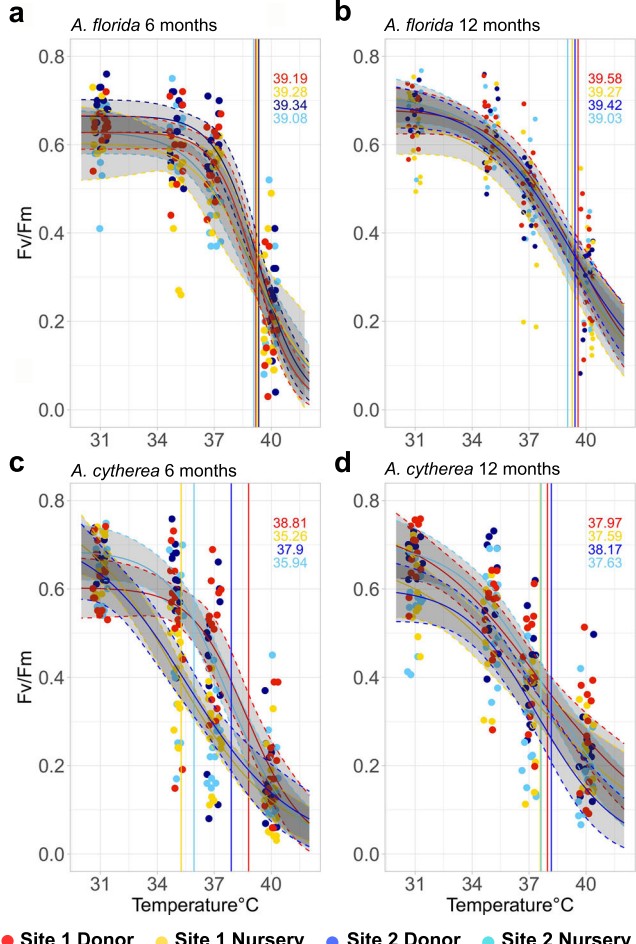

**Fig. 2 | Population-level heat stress dose response curves based on acute heat stress assays (after 7 hours). a, b** Three-parameter log-logistic regression of maximum photosynthetic efficiency of PSII ($F_v/F_m$) over temperature for *Acropora florida* as measured after the heat-hold/ramp down-phase of CBASS assays in (**a**) October 2023 ( ~ 6 months) and (**b**) March 2023 (1 year). **c, d** Three-parameter log-logistic regression of maximum photosynthetic efficiency of PSII ($F_v/F_m$) over temperature for *Acropora cytherea* as measured in (**c**) October 2023 ( ~ 6 months) and (**d**) March 2023 (1 year). The dotted lines represent 95% confidence intervals, and the standard error is indicated by the shaded area based on colony-based curve fits. Vertical lines represent the mean effective dose 50 values (ED50s in °C) of each site-condition group as a proxy for coral thermal tolerance, provided in the top right corner. Dots represent individual data points of the colony replicates.

(www.r-project.org/) using the 'lmerTest' package[61]. Backward elimination was applied to simplify the model's predictor variables (step function), referring to F-tests to compare complete and simplified models, following the removal of fixed effects. Shapiro-Wilk tests were used to test data for normality, and pairwise t-tests (two-tail) were used to compare relative thermal thresholds of individual corals across seasons (i.e., October 2022 and March 2023) within each site-condition group (four groups per species: Site 1-Donor, Site 2- Donor, Site 1-Nursery, and Site 2 -Nursery). Where the assumption of normality was violated, a signed non-parametric Wilcoxon test for paired samples was used instead of a pairwise t-test. Only complete pairs were considered for statistical comparison when sample size differences existed between CBASS assays across seasons. Site-specific differences in relative thermal tolerance thresholds for each species and site-condition group per season (as opposed to across seasons) were tested individually using a Tukey HSD for unequal sample size (two sites, two conditions, a total of four groups per season). Trade-offs between coral growth and relative thermal thresholds were investigated with Pearson's rank correlation coefficient. Hereby, SGRs of nursery corals were measured on day 176 (see

above), and the corresponding ED50 values were ranked per colony from highest to lowest, whereby a significantly negative correlation coefficient would substantiate trade-offs. Growth rate differences between coral species divided into site-condition groups were examined for significant differences with Tukey's post hoc test for multiple comparisons. All data and custom codes detailed and required to reproduce this study are available online[62] and are provided in the supplement (Supplement Data 1).

### Reporting summary
Further information on research design is available in the Nature Portfolio Reporting Summary linked to this article.

## Results
### Species-specific retention vs. recovery of coral thermal tolerance
To assess whether nursery-maintained corals retain, lose, or increase their thermal tolerance in comparison to corresponding reef-based donors in their native habitat, we used the portable Coral Bleaching Automated Stress System (CBASS) and conducted parallel acute heat stress assays at the end of the annual warm season in October 2022 ( ~ 6 months, i.e., 176 days after nursery propagation) and at the end of the colder northeast monsoon season in March 2023 (1 year, i.e., 365 days after nursery propagation). We selected two *Acropora* species from two sites with varying diel temperature environments and disparate reef settings (Fig. 1) to collect coral source material for a common garden nursery at Site 1, established in March 2022. Thermal tolerance of reef-based donor colonies and nursery corals was investigated in parallel by recording dark-acclimated photosynthetic efficiency ($F_v/F_m$) across temperature treatment tanks after the heat-hold/ramp down-phase of the acute heat stress assay thermal cycling profiles for all donor colonies on day 176 (October 2022) and day 365 (March 2023), respectively.

Results from the standardized acute heat stress assays showed species-specific recovery vs. retention of coral thermal tolerance following 365 days of nursery propagation (Fig. 2). Firstly, based on linear mixed effect models, an overall significant difference between nursery corals and the corresponding donor colonies existed for *A. cytherea* (n donor = 52, n nursery = 52, df = 400.02, t = −7.16, p < 0.0001, SE = 0.01) and *A. florida* (n donor = 46, n nursery = 48, df = 371.13, t = −2.65, p = 0.008, SE = 0.01) colonies, considering both reef sites and time points. To examine the overall significant differences in more detail, we conducted pairwise testing between reef-based donor colonies and nursery corals in consideration of the various spatial and temporal scales. This revealed the absence of significant differences between nursery corals and donor colonies for *A. florida* in October 2022 (i.e., day 176) and March 2023 (i.e., on day 365), as indicated by largely similar absolute (i.e., ED50) and relative thermal tolerance thresholds (i.e., ED50-MMM) (Figs. 2, 3, Supplementary Fig. 4), indicating retention of thermal tolerance in nursery settings over a 365-day long nursery period. In contrast, for *A. cytherea* relative thermal tolerance thresholds of nursery corals were significantly lower than donor colonies after 176 days in October 2022 (pairwise t-test p = 0.0006, relative ED50 nursery corals Site 1 = 5.02 ± 0.25 °C vs. relative ED50 donor colonies Site 1 = 7.70 ± 0.52 °C [mean ± SE]; pairwise t-test p = 0.02, ED50 nursery corals Site 2 = 5.66 ± 0.50 °C vs. donor colonies Site 2 = 7.27 ± 0.42 °C, Figs. 2–3). However, these intermediate thermal tolerance differences did not persist. Following the northeast monsoon season in March 2023 (i.e., after 365 days), thermal tolerance thresholds of nursery corals were similar to their reef-based donors (pairwise t-test p = 0.19, relative ED50 nursery corals Site 1 = 7.22 ± 0.46 °C vs. donor colonies Site 1 = 8.23 ± 0.59 °C; Wilcoxon pairwise test p = 0.10, relative ED50 nursery corals Site 2 = 7.09 ± 0.50 °C vs. donor colonies Site 2 = 7.55 ± 0.48 °C) (Fig. 3), suggesting intermediate loss of thermal tolerance followed by subsequent recovery to similar levels as the corresponding reef-based donor colonies.

Through repeated and parallel CBASS testing, spatial and temporal patterns in coral thermal tolerance were investigated to determine the consistency of the thermal response. Although there was a difference in relative thermal thresholds between nursery and donor colonies at the end

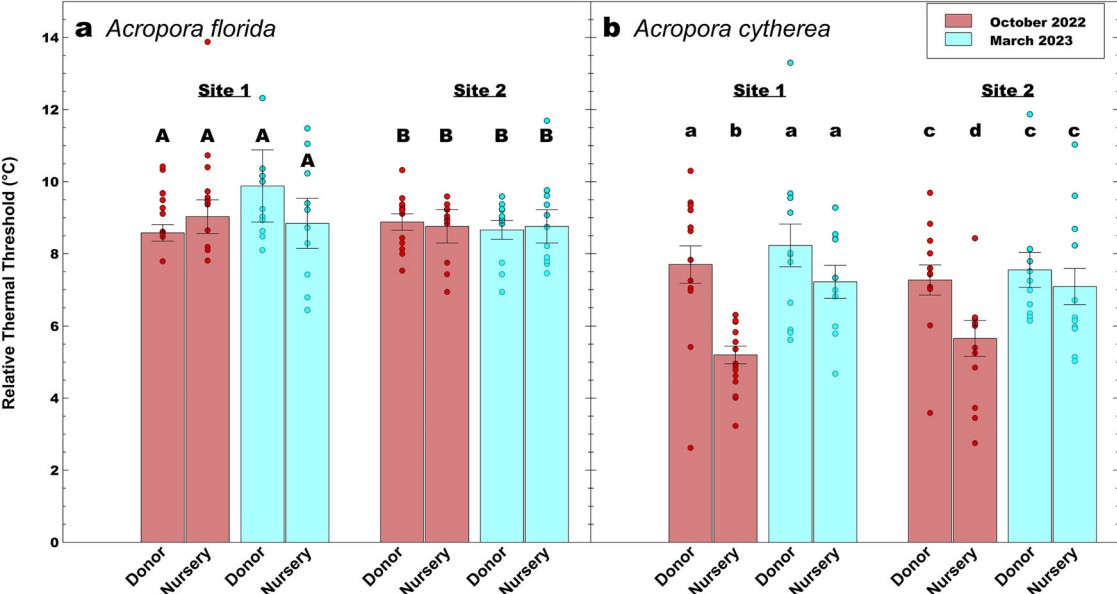

**Fig. 3 | Retention and recovery of coral thermal tolerance across two *Acropora* species.** Colony-specific and population-level thermal tolerance thresholds (mean ± S.E.M) relative to the maximum monthly mean temperature of the study site (relative ED50 = ED50 – MMM; MMM = 30.52 °C based on in situ interpolation of NOAA's climatology data) are shown for each site-condition group (i.e., condition – nursery coral vs. reef-based donor, S1 – site 1, S2 – site 2) of **a** *Acropora florida* and **b** *Acropora cytherea* corals. Annotated letters highlight significant differences (at significance level $p < 0.05$) between groups following pairwise comparison (pairwise two-tail t-tests or signed non-parametric Wilcoxon test, see methods). Dots represent individual coral colony data points.

of the warm season after 176 days in October 2022 for *A. cytherea*, no significant differences between nursery corals across sites were found for both sampling time points, suggesting no effect due to site of origin (Tukey HSD $p = 0.73$, Site 1 October 2022, relative ED50 = 5.02 ± 0.25 °C vs. Site 2 October 2022, relative ED50 = 5.66 ± 0.50 °C and Tukey HSD $p = 0.99$, Site 1 March 2023, relative ED50 = 7.22 ± 0.46 °C vs. Site 2 March 2023, relative ED50 = 7.09 ± 0.50 °C) (Figs. 2, 3, Supplementary Fig. 4). Similarly, no significant differences in relative thermal thresholds of donor colonies across sites could be found (Tukey HSD $p = 0.90$, Site 1 October 2022, relative ED50 = 7.70 ± 0.52 °C vs. Site 2 October 2023, relative ED50 = 7.27 ± 0.42 °C and Tukey HSD $p = 0.88$, Site 1 March 2023, relative ED50 = 8.23 ± 0.59 °C vs. Site 2 March 2023, relative ED50 = 7.55 ± 0.48 °C) (Figs. 2, 3, Supplementary Figs. 4). Finally, no significant differences were determined when comparing relative thermal thresholds of donor colonies across seasons (t-test $p = 0.52$, Site 1 October 2022, relative ED50 = 7.70 ± 0.52 °C vs. Site 1 March 2023, relative ED50 = 8.23 ± 0.59 °C; Wilcoxon pairwise test $p = 0.58$, Site 2 October 2022, relative ED50 = 7.27 ± 0.42 °C vs. Site 2 March 2023, relative ED50 = 7.55 ± 0.48 °C) (Figs. 2, 3). Taken together, the relative thermal thresholds of reef-based donor colonies of *A. florida* were consistent across time and space, while they were consistent across space for *A. cytherea*.

### ED50 reproducibility and rank/cohort assignment consistency
Colony-level reproducibility of ED50 varied among species and conditions (i.e., Donor, Nursery). For *A. florida*, reproducibility was appreciably high as 57% of reef-based donor colonies ($n = 21$) and 62% of nursery corals ($n = 21$) re-tested at an ED50 difference $< \pm 1$ °C when considering ED50 across time points (i.e., October 2022 vs March 2023) (Supplementary Table 4). For *A. cytherea*, ED50 reproducibility was lower, as 50% of reef-based donor colonies ($n = 24$) and 37% of nursery corals ($n = 19$) re-tested at an ED50 difference $< \pm 1$ °C. This variation in ED50 reproducibility between species is further highlighted by the distribution of ED50 values for each site-condition group across time points (Supplementary Fig. 5). By comparison, cohort assignment into top vs. bottom performing colonies showed that ~1 colony was misassigned at a screening size of ≥15 colonies considering the top and bottom five colonies (Fig. 4).

### Coral thermal tolerance and coral growth
To assess putative trade-offs between thermal tolerance and coral growth in a nursery setting following propagation (i.e., putative growth-focused nursery phase), we investigated correlations between ED50s and specific growth rates (SGR) of individual nursery fragments (Table 1, Supplementary Fig. 6). Species-specific coral growth rates after the 176-day nursery period were significantly higher for *A. cytherea* (Table 1). No differences in SGR were found within species across sites, denoting that relocation did not significantly affect growth rates for either species. A negative Pearson correlation of (r) $> -0.30$ was found for *A. cytherea* corals originating from Site 2. No significant correlations between coral thermal tolerance (i.e., ED50) and growth rates were found.

### Discussion
The resilience of coral species to thermal stress is determined by long-term evolutionary processes[63,64] and short-term acclimation[65], which may both contribute equally to coral thermal tolerance[31]. However, and importantly, genotype predisposition to heat stress and coral bleaching remains unaltered when introduced to novel reef environments[32]. For instance, heat-resistant corals retained high metabolic functioning three months after their propagation to novel reef environments in Hawaii[42]. Similarly, in American Samoa and Australia, naturally occurring heat tolerant corals retained their thermal tolerance after relocation to a common garden nursery[40,43]. Taken together, these studies, in line with the results presented here, argue that coral thermal tolerance is adaptively fixed and ultimately retained in novel reef and nursery environments. However, an intermediate acclimation phase following relocation into the nursery may be observed, which can obscure thermal tolerance assessments. Therefore, it is critical to consider the thermal screening of reef-based coral colonies prior to coral nursery stocking, specifically since thermal tolerance is broadly distributed across species, sites, and conditions (Supplementary Fig. 5), suggesting wide thermal tolerance variability. Importantly, we found differences between different species within the same coral genus (i.e., *Acropora*), highlighting that retention vs. recovery of thermal tolerance in novel nursery environments is species-specific[66], and caution should be applied when inferring on a genus basis. In our study, in the absence of repeated heat stress

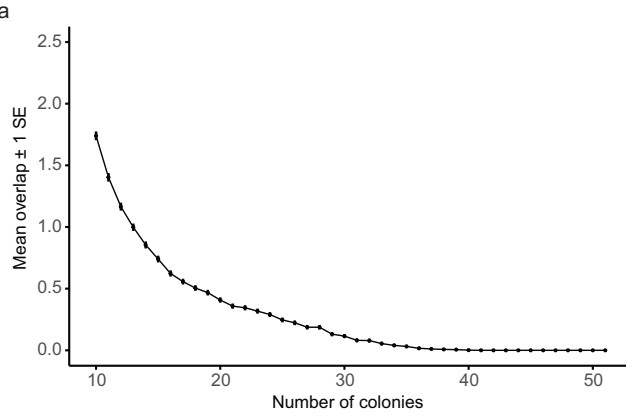

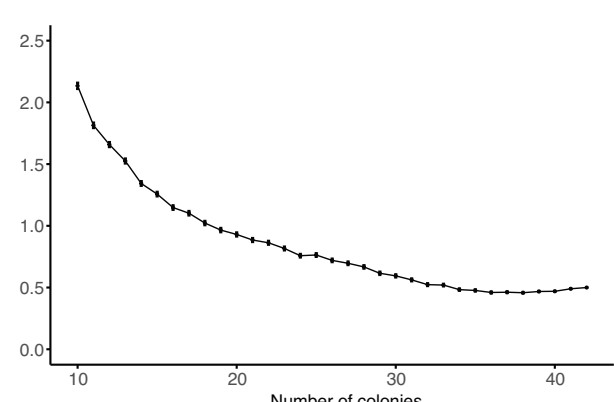

**Fig. 4 | Assessment of assignment consistency into top- and bottom-performing colonies comparing in situ colonies to their nursery counterparts based on screening effort (number of assayed colonies). a** October 2022 and **b** March 2023. Colonies were ranked by their ED50s, subsampled to different sample sizes, and it was assessed whether the top and bottom five colonies were consistent between in situ and nursery corals. Mean overlap indicates the mean number of wrongly assigned colonies. SE = standard error.

### Table 1 | Coral thermal tolerance and growth

| Species | Group | Mean ED50 (°C) | Mean SGR (% growth/day) | Pearson (r) | *p* value |
|---|---|---|---|---|---|
| *Acropora florida* | Site 1 - Nursery | 39.55 | 0.07 a | −0.05 | 0.89 |
| | Site 2 - Nursery | 39.28 | 0.18 a | −0.25 | 0.43 |
| *Acropora cytherea* | Site 1 - Nursery | 35.54 | 0.21 ab | 0.25 | 0.39 |
| | Site 2 - Nursery | 36.18 | 0.28 b | −0.37 | 0.21 |

Pearson's rank correlation analysis of coral thermal tolerance thresholds (ED50s) and specific growth rates (SGRs, % growth/day) of nursery corals after 176 days of growth in a common garden rebar nursery. Letters indicate statistically significant differences in SGR (Tukey HSD multiple comparisons).

experiments over time, results would suggest a considerable loss in thermal tolerance of *A. cytherea* nursery corals in October 2022. Conversely, the initial period of thermal tolerance recovery between March 2022 and October 2022 would not have been noticed by conducting a single heat stress test on day 365 in March 2023, leading to potential implications for restoration practice (as discussed below). Ultimately, and irrespective of the above considerations, we are still missing a definitive understanding to what extent experimentally derived thermal tolerance thresholds map onto ecologically relevant bleaching outcomes[67] regarding susceptibility and ability to recover[68], critical for the implementation of resilience-based coral restoration[69].

Drivers of coral thermal tolerance are complex and diverse. Whereas adaptive differences are fixed and maintained across changing environments[31,42], short-term acclimatory responses may be due to gene expression changes[70] or changes in coral holobiont composition, including Symbiodiniaceae[29,71] and bacteria[33]. Such shifts can also occur following propagation to nursery environments[72], possibly leading to temporary changes in coral thermal tolerance, as observed in this study. Although the specific reasons for the temporary difference in thermal tolerance between nursery corals and corresponding reef-based donors for *A. cytherea* remain undetermined, parallel heat stress testing across time shows that neither seasonality (i.e., site-wide differences in temperature, rainfall, and wind forcing between the warm season and the northeast monsoon season) nor trade-offs between growth and thermal tolerance are explanatory of the difference, as observed by previous studies[47]. Also, discernable differences due to site origin were not detected, but observed differences could be aligned to the species of origin. Thus, metabolomic signatures[73] and protein markers[74], among other variables, may be insightful to further investigate the differences and extent underlying coral thermal tolerance.

Our results suggest that thermal tolerance is adaptively fixed as both assayed species either retained or ultimately recovered their in situ (i.e., site

of origin) thermal tolerance. However, species-specific differences in the form of temporary changes exist. Therefore, an intermediary adjustment period to nursery conditions may be required for some species, whereby coral restoration practitioners must consider the timing of thermal screening and nursery stocking. For example, during severe heat stress in 2019[27], three coral species (i.e., *Acropora muricata*, *Acropora gemmifera*, *Montipora aequituberculata*) in suspended rebar nurseries at this common garden site showed a higher average bleaching response than their corresponding reef-based donors, 90 days after coral nursery propagation[54]. However, no differences were found during a heat stress period in the subsequent year, suggesting and confirming recovery of initial (i.e., in situ) thermal tolerance following long-term rearing in the nursery[40]. Furthermore, while thermally more resilient phenotypes retained their tolerance in a common garden nursery in American Samoa, coral bleaching was induced at milder temperatures in the multi-species nursery compared to temperatures experienced by corresponding donor colonies in their native habitats[43], possibly due to the short time frame elapsed since nursery stocking (i.e., approximately six months).

Taken together, corals can maintain and recover their stress response physiology[75,76] but maximizing time in the nursery before potential periods of heat stress occur will be crucial to ensure that identified heat tolerant coral genotypes recover their original thermal tolerance (as measured for reef-based donor colonies). In addition, while the capacity of acute thermal screening (CBASS'ing) to reliably identify resilient phenotypes must be verified in situ, e.g., by following bleaching trajectories of diverse coral colonies in the field[37,67,77], this study adds to previous research signifying the importance of preferential selection of thermally resilient coral phenotypes for restoration[40–43,78]. Pre-screening of coral phenotypes (i.e., acute heat stress testing to identify corals with superior thermal tolerance) should ideally occur before nursery stocking by testing a high number of reef-based donor colonies to safeguard coral nursery survival and restoration

investment. Such efforts should further be accompanied by parallel genotyping of selected coral individuals to prevent genetic depression of nursery populations[79] and to ensure cross-tolerance to other pervasive stressors[44,69]. Importantly, the reproducibility of colony-level ED50 is critical for restoration application to confidently select heat-tolerant corals. Although the ED50 reproducibility here was lower compared to Cunning et al., 2024 (i.e., 37–62% vs. 90% of corals re-tested at ED50 difference of < ±1 °C)[60], the observed rates are still appreciably high, specifically when considering the thermal tolerance loss and recovery of *A. cytherea* nursery corals and known seasonal adjustments of ED50s of > 1 °C for *Acropora* species in the Red Sea[80]. For real-world application, consistency of cohort assignment (higher vs. lower thermal tolerance) may ultimately become more important than high reproducibility of ED50s. Our results show that top and bottom cohort assignment is highly consistent between reef-based donor and nursery corals for both time points, with the October 2022 time point showing higher consistency than the March 2023 time point. This may be attributable to differences in thermal tolerance threshold recovery, seasonal acclimation, etc. At large, our results suggest that with a reasonable screening effort of 20 colonies and subsequent selection of the top performing colonies, high accuracy can be achieved, emphasizing the suitability of CBASS assays to differentiate between heat tolerant and susceptible colonies across reef sites and species. However, at this conjecture, it is vitally important to resolve how ED50 differences map onto ecological in situ bleaching outcomes. This requires investigations of coral thermal tolerance and resilience to prevailing stressors, which must be conducted at a species and population level across large sample sizes and habitats. Importantly, discernable patterns of coral thermal retention and recovery are shown for two widely distributed *Acropora* species sourced from the same sites, reared in a common garden nursery, and subjected to equal heat loads during CBASS experiments. Thus, extending insights from one species to other species (even) of the same genus may undermine future research and restoration efforts.

## Conclusion

Our study provides evidence that coral thermal tolerance is adaptive and retained in nursery settings, but discernible species-specific differences exist, whereby some species may require an adjustment period in common garden nurseries before original thermal tolerance is recovered. Due to these differences in two congeneric *Acropora* species, it is critical to conduct experiments at the species level without inferring results from one species to another, even if they are of the same genus. Results from this study imply that the identification of heat tolerant corals through heat stress screening of coral source material should ideally happen prior to the nursery phase by assaying reef-based donor colonies across diverse reef habitats. Similarly, coral nurseries should be stocked in foresight of heat stress periods to allow for sufficient time for corals to recover their original thermal tolerance. Future studies should determine changes in coral physiology and holobiont assemblage across coral restoration stages to identify drivers underlying temporary changes in thermal tolerance. Critically, in situ validation of experimentally derived thermal thresholds is required across reef scales at the species level to determine the capacity of acute thermal assays to reliably identify corals with increased thermal resilience under ecologically (i.e., real-world) relevant conditions for further application in coral restoration practice.

## Data availability

All data required to perform the analyses and generate the results and figures in this manuscript are available in the Supplement Data 1 and are deposited open-access in the Zenodo repository: https://doi.org/10.5281/ZENODO.15880518.

## Code availability

Statistical software codes and detailed data required to reproduce this study are available in the supplement as well as in the Zenodo repository: https://doi.org/10.5281/ZENODO.15880518.

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

## Acknowledgements

The research was conducted under permit number Prk. ML. 630-7Jld.12 (90), issued by the Department of Fisheries (DoF) Malaysia (Jabatan Perikanan Malaysia), and permit number EPU 40/200/19/3711 (13) issued by the Economic Planning Unit (Unit Perancang Ekonomi). We would like to thank research assistants Natasha Zulaikha, Mok Weng Dee, and Febrianne Sukiato for assisting with data collection and acute heat stress assays, and Affendi Yang Amri for providing us with the underwater pulse amplitude modulated (PAM) fluorometer to conduct the study. Special gratitude is extended to Lush Malaysia for funding the experimental systems by awarding a Lush Charity Pot grant to Coralku Solutions, and to Summer Bay Resort and the dive center team for supporting our fieldwork actively and in-kind.

## Author contributions

Study design—S.S., C.R.V.; coral sampling and nursery stocking—S.S.; data collection—S.S., C.K.L.; formal analysis—S.S., C.R.V.; Original draft-S.S.; writing and revisions—S.S., C.R.V., C.K.L.

## Funding

## Competing interests

The authors declare no competing interest.
