## [Transparent Peer Review file · Communications Biology]

Species-specific retention vs. recovery of coral thermal tolerance following nursery propagation

Corresponding Author: Mr Sebastian Szereday

Version 0:

Reviewer comments:

Reviewer #1

(Remarks to the Author)

The authors compare the thermal tolerance of donor colonies and nursery reared fragments of *Acropora cytherea* and *Acropora florida* sourced from 2 sites in Malaysia. They found some species-specific responses after 6 months, but overall thermal tolerance of nursery-reared colonies were similar to that of donor colonies after 1 year. The research question posed in this study is novel and will be of interest to researchers examining coral thermal tolerance as well as reef restoration managers.

I do, however, have some concerns about the validity of the conclusions drawn by the authors. The main concern is the lack of replication among CBASS temperature treatments (based on my understanding there was only $n=1$ tank per temperature for each time point). This is in contrast to the protocols outlined in the original CBASS design, which included a least one replicate tank per temperature treatment. The lack of replication in experimental design merits caution in the interpretation/confidence in the results presented. That being said, I appreciate the author's inclusion of 2 time points and otherwise their sample size and study duration are sufficient. Yet, on a fundamental experimental design level I am not sure if the similarities/differences among time points are biologically driven or are simply a function of variation amplified by the lack of replication. I am not suggesting that this study needs to be repeated or that it isn't suitable for publication, but the authors need to include a clear statement about the lack of replication and how it could influence their overall findings.

Additionally, authors need to clearly explain why some nursery fragments were not included and how this may have affected the study findings (for this I am referring to the subset included in the original sampling time point, not the second time point for which the authors have been clear about the reasons for excluding some donor/nursery fragments). Lastly, the authors should provide access to their R code on the GitHub link for transparency.

After revisions are made, I think this manuscript will be suitable for publication in *Communications Biology*. I wish the authors all the best with their revision, and I do appreciate the hard work that went into this study.

Abstract

Line 16-17: "is currently considered" seems quite vague; perhaps add a bit more context to this sentence

Line 21: I do not think 'offspring' is appropriate here; these are simply fragments of the donor colonies, not offspring. Please revise wording.

Introduction

Overall, the introduction is well-written and supported with relevant/current literature. One aspect that may add context to the introduction, however, would be to expand on the risks/drawbacks of selecting donor colonies based on thermal performance alone (this is touched on slightly at the end of the introduction, but for readers that are not directly familiar with the studies in the references there may be a lack of recognition of this consideration).

Methods

Line 98: Please specify that the nursery is in situ

Line 150: Why were only a subset of the original fragments used? Was there mortality in other fragments? Or alternatively, had a subset of fragments not grown as efficiently (i.e., not enough material available to make enough fragments for the CBASS?). How might this have affected the results (both in the context of the exclusion of these nursery corals and their donor colonies in the authors analysis)? This needs to be clearly stated and addressed for transparency.

General comment: While the CBASS methods are clearly outline in other papers, the authors here need to be clear about their lack of replication across temperature treatments (n=1 tank/temperature/time point) and how this may have influenced their findings. How confident are the authors that the differences across time points in thermal tolerance in *A. cytherea* are due to biological rather than experimental design influences?

Data availability: Raw data is clearly given on the GitHub link, but I do not see any R script available. In the spirit of transparency (and also to meet journal guidelines), it would be best if this was also provided.

Results

Line 239-250: This paragraph is better suited for the methods section.

Line 298: Can the authors include some information about the Symbiodiniaceae type at each site (while I understand this was not included in their study, can they reference other papers that provide this information).

Discussion

Line 354: I do appreciate the authors' inclusion of more than one time point. This is an important consideration that requires more attention in coral thermal tolerance studies.

Line 402: Can you include some field-based CBASS studies here (if any)?

General comment: This study only looked at 2 main response parameters (Fv/Fm and growth). The authors need to acknowledge that there may be differences among donor colonies/nursery fragments in other parameters that were not assessed in this study.

Figures & Tables

Figure 1: Please provide more context on the zoomed-out figure (e.g., closest major city name)

Figure 2a: Why is there a white box covering some of the graph?

Reviewer #2

(Remarks to the Author)

Dear authors,

Congratulations on an excellent study and a well-written manuscript. I have only a few minor comments aimed at clarifying specific points, but overall, I believe this work makes a valuable contribution to the field of coral restoration.

Your study convincingly demonstrates through a CBASS heat stress experiment that the photosynthetic performance of two *Acropora* species is maintained following nursery rearing. While one species exhibited a significant drop in performance after the first few months, this decline was temporary, and recovered after a year, bringing performance levels back in line with those of the donor colonies.

The study is elegant, thoughtfully designed, and clearly presented. The conclusions are well-supported by the data and offer meaningful insights for coral restoration practices.

Reviewer #3

(Remarks to the Author)

In this study Szereday and co-authors present data on how heat tolerance is retained in a nursery setting from two reef-building coral species of restoration importance. They found that *A. cytherea* heat tolerance means were lower for nursery colonies compared to donor colonies at their first time point (~6 months after transplant) but not the second time point (1 year after transplant). They do not observe these differences between conditions in *A. florida*. These results suggest species-specific differences in the timing of acclimation to nursery habitat, with the important message to restoration practitioners that heat tolerance screening should take place prior to transplant to nurseries. This work is of importance to the community and should be considered for publication following minor revisions below.

Version 1:

Reviewer comments:

Reviewer #1

(Remarks to the Author)

The authors have addressed my comments and I believe the manuscript is suitable for publications. Congrats to the authors!

Reviewer #1 (Remarks to the Author):

The authors compare the thermal tolerance of donor colonies and nursery reared fragments of *Acropora cytherea* and *Acropora florida* sourced from 2 sites in Malaysia. They found some species-specific responses after 6 months, but overall thermal tolerance of nursery-reared colonies were similar to that of donor colonies after 1 year. The research question posed in this study is novel and will be of interest to researchers examining coral thermal tolerance as well as reef restoration managers.

Response: Thank you for your review. We appreciate the overall acknowledgment of the importance and novelty of our work. Below we provide responses to your specific comments, detailing how we addressed them in the manuscript revision.

I do, however, have some concerns about the validity of the conclusions drawn by the authors. The main concern is the lack of replication among CBASS temperature treatments (based on my understanding there was only n=1 tank per temperature for each time point). This is in contrast to the protocols outlined in the original CBASS design, which included at least one replicate tank per temperature treatment. The lack of replication in experimental design merits caution in the interpretation/confidence in the results presented. That being said, I appreciate the author's inclusion of 2 time points and otherwise their sample size and study duration are sufficient. Yet, on a fundamental experimental design level I am not sure if the similarities/differences among time points are biologically driven or are simply a function of variation amplified by the lack of replication. I am not suggesting that this study needs to be repeated or that it isn't suitable for publication, but the authors need to include a clear statement about the lack of replication and how it could influence their overall findings.

Response: Thank you for bringing this up. In the original CBASS design (Voolstra et al. 2020), we indeed used replicate tanks with replicated fragments that did not produce a significant effect, which then provided the rationale to only consider values from a single replicate. The original text from Voolstra et al. (2020) reads:

“Initial analysis of PAM fluorometry data revealed *no significant effect of replicated tanks* either for CBASS ($\chi^2 (1) = 0.418, p = .517$) or CLASSIC ($\chi^2 (1) = 2.84 \times 10^{-14}, p = 1$) using the function *ranova*. Hence, for all other measurements, two-way ANOVAs sum of squares type III were conducted with temperature treatment and reef site as fixed factors *on samples from a single replicate of tanks* in order to save sample processing time.”

In some subsequent studies, we still used replicated tanks/fragments, although eventually we transitioned to the rationale that using single fragments per temperature treatment is sufficient. This is broadly for three reasons: i) we did not measure significant tank effect differences (see above); ii) we aim to limit biomass harvesting from coral reefs (using single replicates, effectively halves the impact); iii) biological replicates entail aspects of technical variance, thus increasing number of biological replicates seems more insightful. Arguably, this does not address the case that a given tank would

malfunction. However, in this scenario all fragments from that tank would be affected equally and this would be noticed.

We have added text to the Materials & Methods section in the revised manuscript to address your concern and the above considerations.

Text reads:

“Of note, we did not use replicated fragments (tanks) for each temperature treatment, since a prior study highlighted the absence of significant tank effects (Voolstra et al. 2020) and to minimize biomass harvesting. Further, we assayed a high number of coral colonies (n = 12 for A. florida and n= 14 for A. cytherea) to capture biological variation, which also encompasses aspects of technical variance inherent to the experimental workflow.”

Additionally, authors need to clearly explain why some nursery fragments were not included and how this may have affected the study findings (for this I am referring to the subset included in the original sampling time point, not the second time point for which the authors have been clear about the reasons for excluding some donor/nursery fragments).

Response: Thank you. This point is addressed in detail in our response below.

Lastly, the authors should provide access to their R code on the GitHub link for transparency.

Response: Of course! This was an oversight on our behalf. For the revised version, the GitHub repository link is provided. <https://github.com/Coralku/commongarden>

After revisions are made, I think this manuscript will be suitable for publication in Communications Biology. I wish the authors all the best with their revision, and I do appreciate the hard work that went into this study.

Response: Thank you.

Abstract

Line 16-17: “is currently considered” seems quite vague; perhaps add a bit more context to this sentence

Response: Thank you. We have revised the sentence to explicitly refer to thermal pre-screening.

Revised sentence reads:

“Thermal screening of coral source material to enhance long-term restoration success under ocean warming may be crucial.”

Line 21: I do not think ‘offspring’ is appropriate here; these are simply fragments of the donor colonies, not offspring. Please revise the wording.

Response: Revised accordingly following the updated coral restoration terminology detailed in Suggett et al. 2025 - *A user’s guide to coral reef restoration terminology* (<https://doi.org/10.1007/s00338-025-02619-8>)

Revised text is underlined and reads:

*“Here, we used CBASS acute thermal assays to compare standardized thermal tolerance thresholds (ED50s) of donor colonies from *Acropora cytherea* and *Acropora florida* from two sites in Pulau Lang Tengah, Malaysia, to their ‘nursery propagule’ reared in a common garden coral nursery over 365 days.”*

Introduction

Overall, the introduction is well-written and supported with relevant/current literature. One aspect that may add context to the introduction, however, would be to expand on the risks/drawbacks of selecting donor colonies based on thermal performance alone (this is touched on slightly at the end of the introduction, but for readers that are not directly familiar with the studies in the references there may be a lack of recognition of this consideration).

Response: We agree, this is an important consideration, and such risks may exist (or are unknown currently) for some species. We included this in the introduction and included relevant literature text. The text reads as follows:

*“Although preferential selection of heat tolerant corals for restoration is considered crucial under warming oceans (Caruso et al., 2021), this may come with associated risks. For example, compromised growth and reduced survivorship of outplanted thermally tolerant corals have been noted (Ladd et al., 2017; Ho‘opai-Sylva et al., 2025). Such trade-offs may stem from associations with thermally tolerant algal symbiont types (Jones and Berkelmans, 2010). However, contrasting evidence exists. Co-benefits (i.e., positive associations) between physiological performance traits such as fecundity, growth, and thermal tolerance were found in corals from the Indo-Pacific (Lachs et al., 2023b), while cross-tolerance to multiple stressors have been identified for the common Indo-Pacific *Acropora millepora* (Wright et al., 2019). Thus, trade-offs associated with thermal tolerance may not be universal across species or may not universally manifest across environments (Rivera et al., 2022).”*

Methods

Line 98: Please specify that the nursery is in situ

Response: Revised accordingly.

Line 150: Why were only a subset of the original fragments used? Was there mortality in other fragments? Or alternatively, had a subset of fragments not grown as efficiently (i.e., not enough material available to make enough fragments for the CBASS?). How might this have affected the results (both in the context of the exclusion of these nursery corals and their donor colonies in the authors analysis)? This needs to be clearly stated and addressed for transparency.

Response: Thank you, this is a very good point, and it is correct that we have to further describe the selection of nursery corals for CBASS assays. We omitted doing so, due to several reasons that led us to conclude that the haphazard selection of nursery corals did not bias the outcomes of the study.

Our target sample size was 12-14 corals per species for each Site and Condition (e.g., Nursery Site 1, Donor Site 1, etc), which corresponds to the maximum number of colonies/fragments to be run in a CBASS assay. To achieve this, we overstocked the nurseries to counter eventual mortality. Indeed, the haphazard selection of nursery corals underwater was conducted after some level of mortality affected the coral nurseries, so not all corals were available for experimentation. Fortunately, mortality was limited, and sufficient corals survived to achieve the target sample size, and we were able to select corals haphazardly from the surviving nursery pool.

Importantly, the corals within this surviving nursery pool included nursery corals with good, bad, and no growth. This is reflected in the growth data presented in Table 1 and ESM8 (data points close to 0% growth per day). For selected *A. florida*, the SGR ranged from -0.12 to 0.34 %growth/day; for *A. cytherea*, this range was 0.07-0.49 %growth/day. The ED50 vs. SGR comparison therefore captured corals across the growth spectrum. However, we note that the corals sampled were on average larger than the non-sampled corals, to ensure sufficient biomass would be available six months later in March 2023 for the second round of CBASS assays. This is now stated in the revised text (underlined):

*“After coral growth and survival monitoring on day 176 in October 2022, a subset of the total sample size was selected to achieve the maximum sample size of n=14 per CBASS assay. Hereby, 24 nursery corals of *A. florida* (n=12 from both sites) and 27 nursery corals of *A. cytherea* (n=14 from Site 1, n=13 from Site 2) were selected for acute heat stress testing together with their reef-based donor counterparts, representing 24 unique *A. florida* and 27 *A. cytherea* reef-based donor colonies. A total of 192 *A. florida* coral nubbins (i.e., 24 reef-based donors x 4 fragments per colony + 24 nursery corals x 4 fragments per colony) and 216 *A. cytherea* nubbins (27 reef-based donors x 4 fragments per colony + 27 nursery corals x 4 fragments per colony) were collected. The selection of the colony subsets was conducted haphazardly and included fragments of varying colour intensity (i.e., from slightly pale to fully pigmented) of variable growth rates (i.e., 0.12-0.34 %growth/day for *A. florida*, and 0.07-0.49 %growth/day for *A. cytherea*). Of note, selected nursery corals for CBASS assays were on average larger than non-selected corals to ensure sufficient coral biomass for subsequent CBASS assays in March 2023.”*

General comment: While the CBASS methods are clearly outlined in other papers, the authors here need to be clear about their lack of replication across temperature treatments (n=1 tank/temperature/time point) and how this may have influenced their findings. How confident are the authors that the differences across time points in thermal tolerance in *A. cytherea* are due to biological rather than experimental design influences?

Response: Please refer to our response above. Thank you.

Data availability: Raw data is clearly given on the GitHub link, but I do not see any R script available. In the spirit of transparency (and also to meet journal guidelines), it would be best if this was also provided.

Response: Thank you. Codes and scripts needed to reproduce the study results have now been uploaded. GitHub repository link:

<https://github.com/Coralku/commongarden>

Results

Line 239-250: This paragraph is better suited for the methods section.

Response: Thank you. We internally discussed this. Given that many readers jump to the results with little consideration of methodology, we eventually decided that this paragraph fits well here to provide context. We hope you agree.

Line 298: Can the authors include some information about the Symbiodiniaceae type at each site (while I understand this was not included in their study, can they reference other papers that provide this information).

Response: We agree, this would be a welcomed input to contextualize the study, but unfortunately, such data are unavailable for this region. Current research efforts are investigating symbiotic associations of multiple species from the study sites, including *A. cytherea* and *A. florida*.

Discussion

Line 354: I do appreciate the authors' inclusion of more than one time point. This is an important consideration that requires more attention in coral thermal tolerance studies.

Response: Thank you for the positive feedback. We agree, it is crucial to consider multiple time points and seasonality, as this paper shows as well: <https://doi.org/10.1038/s42003-024-07340-w>

Line 402: Can you include some field-based CBASS studies here (if any)?

Response: We included Brown et al., 2023 - Divergent bleaching and recovery trajectories in reef-building corals following a decade of successive marine heatwaves (<https://doi.org/10.1073/pnas.2312104120>), and Evensen et al. 2022 (<https://doi.org/10.1007/s00338-022-02233-y>) as relevant field-based studies (in addition to those cited at the end of this sentence, i.e., Morikawa and Palumbi, 2019; Alderdice et al., 2024). Notably, we intend to refer to studies conducting CBASS *prior to* in situ bleaching events to verify the predictive skill of CBASS, which are unavailable at this moment (although we are currently finalising a manuscript on this topic).

General comment: This study only looked at 2 main response parameters (Fv/Fm and growth). The authors need to acknowledge that there may be differences among donor colonies/nursery fragments in other parameters that were not assessed in this study.

Response: Agreed, there are several other parameters, and we have added text at L446-450 to acknowledge protein and metabolomic signatures:

“Although this study demonstrates a clear difference in photochemical yields (i.e., Fv/Fm) between reef-based donor colonies and nursery corals after six months, metabolomic signatures (Roach et al., 2021) and protein markers (Nunn et al., 2025), among other variables, may be insightful to further investigate the differences and extent underlying coral thermal tolerance.”

Figures & Tables

Figure 1: Please provide more context on the zoomed-out figure (e.g., closest major city name)

Response: We added a location pin to highlight the location of Malaysia’s capital city Kuala Lumpur and added longitude and latitude to the map.

Figure 2a: Why is there a white box covering some of the graph?

Response: Thank you for identifying this error. We have removed the white box.

Reviewer #2 (Remarks to the Author):

Dear authors,

Congratulations on an excellent study and a well-written manuscript. I have only a few minor comments aimed at clarifying specific points, but overall, I believe this work makes a valuable contribution to the field of coral restoration.

Your study convincingly demonstrates through a CBASS heat stress experiment that the photosynthetic performance of two *Acropora* species is maintained following nursery rearing. While one species exhibited a significant drop in performance after the first few months, this decline was temporary, and recovered after a year, bringing performance levels back in line with those of the donor colonies.

The study is elegant, thoughtfully designed, and clearly presented. The conclusions are well-supported by the data and offer meaningful insights for coral restoration practices.

Response: Thank you for the praise! We provide responses to your specific comments below, detailing how we addressed them in this revision.

Fig1 – can you at Lat Long coordinates to the map?

Response: Fig1 has been revised accordingly, and coordinates were added.

L119-121: The phrase "kept constant" is a bit unclear. I think you mean that the depth of the frame was kept at the same level as the average depth of the donor colonies. Maybe rephrase it to just articulate that a little clearer?

Response: Thank you, we have rephrased this accordingly.

Revised text reads:

"The average depth of rebar frames was 9.0 m (i.e., middle point of the frame, top = 8.8 m, bottom 9.2 m) and was kept at the same depth level as the average depth of reef-based donor colonies (i.e., 9.2 m)"

L126: How are length and height different? I understand you measured width, but the difference between length and height is unclear to me.

Response: Length is the maximum horizontal linear extension of the coral fragment, while height is the maximum vertical linear extension. In a nutshell, we measure extension along all three spatial axes (x, y, z), which we refer to as length, height, and width, respectively, for consistency with common usage and intuitive interpretation.

Line 148: Could you provide more ecological context for the chosen temperature treatments? Is 39.5°C a temperature these corals are actually exposed to during an extreme heatwave, or is it intended to push the corals thus far to see a drop in their performance? It seems extremely high. Adding a brief explanation would help readers better interpret the relevance of the results.

Response: Correct, the treatment temperatures are meant to induce heat stress at variable levels, whereby the highest temperature is intended to ‘push coral over the edge’. This ensures a more robust ED50s estimate, as accurate log-logistic modelling requires the curve to ‘bend’. From an ecological point of view, these temperatures are extreme. We added a brief description to clarify this at L160-163:

“Heat-hold temperatures were chosen to cover the spectrum of physiological responses to heat stress (from mild to extreme) for subsequent log-logistic modelling of photosynthetic efficiency loss over increasing temperatures, whereby the highest temperature is intended to induce severe bleaching and mortality (Voolstra et al., 2025)”

Lines 149–155: I'm having some trouble following how the fragments were sampled. The “n=xxx” for each species is clear, but it would be helpful to know how many donor colonies this represents. For example, line 154 mentions four replicate fragments per donor colony—does this mean you sampled from three donor colonies (with four fragments each) to reach $n=12$ for *A. florida*? Clarifying this would make the sampling design easier to follow.

Response: Thank you for highlighting this. We added some wording to clarify this. To be explicit, 24 unique donor colonies and their nursery counterparts were tested for *A. florida*. For each time point, 4 fragments from each colony were collected to run 4 different thermal profiles for each CBASS assay. This comes up as follows: 24 (unique donors) x 4 (fragments per donor) + 24 nursery corals x 4 = 192 coral nubbins were used for CBASS testing for *A. florida*. For *A. cytherea*, this number is 27 x 4 + 27 x 4 = 216 coral nubbins (please see L169-180)

Revised text reads:

*“After coral growth and survival monitoring on day 176 in October 2022, $n=14$ colonies were assayed, corresponding to the maximum number of colonies that can be run in a single CBASS assay. Hereby, 24 nursery corals of *A. florida* ($n=12$ from each site) and 27 nursery corals of *A. cytherea* ($n=14$ from Site 1, $n=13$ from Site 2) were selected for acute heat stress testing together with their reef-based donor counterpart, representing 24 unique *A. florida* and 27 unique *A. cytherea* reef-based donor colonies. A total of 192 *A. florida* coral nubbins (i.e., 24 reef-based donors x 4 fragments per colony + 24 nursery corals x 4 fragments per colony) and 216 *A. cytherea* nubbins (27 reef-based donors x 4 fragments per colony + 27 nursery corals x 4 fragments per colony) were collected and assayed.”*

Lines 182–188: Am I right in understanding that you re-measured the same nursery fragments, but had to sample new fragments from the donor colonies at the sites? If not, how did you maintain the original fragments for re-measurement? A bit more explanation here would help clarify the methodology.

Response: We did not re-measure the same nursery fragments. Rather, just like re-sampling the donor colonies in situ, we collected new fragments from each previously tested nursery corals. This has been clarified in the text.

Revised text reads:

“In March 2023, 365 days after the first nursery stocking, standardized heat stress assays were again conducted to assess the consistency of coral thermal tolerance. For this, previously tested nursery corals and reef-based donor colonies were both resampled for acute heat stress testing, and new fragments from each previously tested nursery and reef-based donor were collected for subsequent CBASS assays.”

L206-208: I think you should start this paragraph with this definition of “coral thermal tolerance thresholds”, rather than stating this halfway through this paragraph.

Response: Thank you. Revised accordingly.

Revised text reads (changed underlined):

“Coral thermal tolerance thresholds (i.e., Effective Dose 50 - ED50s) are defined as the mean temperature at which 50% of the value measured (here: F_v/F_m) at baseline temperature is retained (Voolstra et al., 2025). For quantitative comparison of species-specific standardized thermal tolerance thresholds between nursery corals and reef-based donors,”

Statistical analyses look robust and good to me although I would not consider myself an expert in this domain.

Response: Thank you.

Results

Fig2 – there is a little white square in panel a at x=40 degrees. Is this a formatting error?

Response: Thank you for spotting this. We have removed the white box, which was there in error.

How did you calculate the relative thermal threshold? I can't seem to find this in the methods. I read about how you got to ED50 but not the relative thermal threshold, which is needed to understand Fig3.

Response: Thanks. Upon revisiting this, we agree that clarity can be improved.

Revised text reads:

“To calculate relative thermal tolerance thresholds, we subtracted the MMM from the determined ED50s to obtain relative thermal thresholds (ED50 – local MMM). This was done separately for each site and condition, i.e. Site 1 - Donor, Site 2 - Donor, Site 1 - Nursery, and Site 2 - Nursery.”

Discussion

L. 405–406 and 422–423: I'm unclear on the concept of pre-screening phenotypes through heat stress. As I understand it, your study did not conduct pre-screening—rather, the heat stress test was done after >150 days, and only then was a drop in Fv/Fm observed in one species. So what exactly is the pre-screening for? Is it to understand that although performance may dip temporarily it should return to pre-stocking/pre-screening levels? If so, this could be explained a little more clearly. Mostly, it would be helpful to better understand the purpose of pre-screening (I guess answer a little clearer the “so what” question).

Response: Thank you for this suggestion. We added text to improve clarity. By pre-screening we are referring to heat stress testing of reef-based donor colonies across reef habitats, without prior knowledge of their thermal tolerance. In the real world, this would enable the selection of corals with superior thermal tolerance for restoration purposes (see also: <https://www.nature.com/articles/s41598-024-51944-5>). The text has been revised to highlight this conclusion.

Revised text is underlined:

“Pre-screening of coral phenotypes (i.e., acute heat stress testing to identify corals with superior thermal tolerance) should ideally occur before nursery stocking by testing reef-based donor colonies to safeguard coral nursery survival and restoration investment”

and:

“Results from this study imply that the identification of heat tolerant corals through heat stress screening of coral source material should ideally happen prior to the nursery phase by assaying reef-based donor colonies across diverse reef habitats.”

Reviewer #3 (Remarks to the Author):

In this study Szeready and co-authors present data on how heat tolerance is retained in a nursery setting from two reef-building coral species of restoration importance. They found that *A. cytherea* heat tolerance means were lower for nursery colonies compared to donor colonies at their first time point (~6 months after transplant) but not the second time point (1 year after transplant). They do not observe these differences between conditions in *A. florida*. These results suggest species-specific differences in the timing of acclimatization to nursery habitat, with the important message to restoration practitioners that heat tolerance screening should take place prior to transplant to nurseries. This work is of importance to the community and should be considered for publication following minor revisions below.

Response: Thank you for the acknowledgement of the importance of the work and for the suggestion on how to further improve things. We provide responses to your specific comments below, detailing how we addressed them in the revised manuscript.

Lines 16-17: Revise this sentence as is currently awkwardly worded e.g., “...to enhance restoration success is being considered by/for...”

Response: Thank you. We have revised the sentence to explicitly refer to thermal pre-screening.

Revised sentence reads:

“Thermal screening of coral source material to enhance long-term restoration success under ocean warming may be crucial.”

Lines 20-21: I understand what’s intended by “nursery offspring” here but I’m concerned it may be misinterpreted as actual offspring.

Response: Thank you, agreed. Revised accordingly following the updated coral restoration terminology detailed in Suggett et al. 2025 - *A user’s guide to coral reef restoration terminology* (<https://doi.org/10.1007/s00338-025-02619-8>)

Revised text reads:

*“Here, we used CBASS acute thermal assays to compare standardized thermal tolerance thresholds (ED50s) of donor colonies from *Acropora cytherea* and *Acropora florida* from two sites in Pulau Lang Tengah, Malaysia, to their ‘nursery propagule’ reared in a common garden coral nursery over 365 days.”*

Line 45: And importantly marine heatwaves are becoming more frequent

Response: Thank you, that is correct. Revised to include this point.

“However, concurrently with advances in coral restoration practice, ocean warming is resulting in longer, more widespread and more frequent marine heatwaves (Skirving et al., 2019; Eakin et al., 2022), resulting in severe mass coral bleaching and mortality events in all ocean basins (Hughes et al., 2018b; Eakin et al., 2022; Reimer et al., 2024).”

Line 61: What is meant by “markers” to identify heat tolerant phenotypes? This suggests biomarkers or genetic markers which I wouldn’t consider “globally advanced”.

Response: Yes, we were referring to biomarkers (e.g., molecular signatures, such as diagnostic ITS2 and 16S sequences, or expressed proteins). However, we agree that these are not globally advanced as the text states. Thus, the text here does not apply to biomarkers, and we removed this word from the text.

Revised text reads:

“Despite the complex underpinnings of coral thermal tolerance, recent global advances have been achieved to develop standardized diagnostic tests to identify coral phenotypes with superior heat stress tolerance (Voolstra et al., 2020, 2021b, 2025; Evensen et al., 2023)”

Lines 359-362: See Caruso et al 2025 (“Short-term stress testing predicts subsequent natural bleaching variation”) for an update here, though your statement holds on how we are still missing a definitive understanding.

Response: Thank you. We included this study to the statement.

General results:

Was it unexpected that the 2 sites did not differ in their heat tolerance, given their environmental differences?

Response: No, this was not an unexpected outcome. Although temperature regimes across these sites are significantly different, and variability across sites exists in terms of wind and wave exposure, differential in situ bleaching responses were not noted for *Acropora* corals during the most recent bleaching events in 2019-2020 (Szereday et al., 2024 – <https://doi.org/10.1007/s00227-024-04495-2>). However, we expected nursery corals from Site 1 to potentially perform better during CBASS testing than corals sourced from Site 2, since Site 1 nursery corals were not ‘site relocated’.

It looks like there was a wide variation in thermal tolerances within species/sites/conditions. As your study directly relates to choosing individuals for restoration, it would be interesting to see the distribution of these colony ED50s to

understand if certain colonies/genotypes may be more affected than others by transplantation to the nursery. Also, was there a correlation in ED50s over time (as seen in Cunning et al “Census of heat tolerance among Florida’s threatened staghorn corals finds resilient individuals throughout existing nursery populations” Figure 5)?

Response: Thanks. Indeed we find a broad distribution of thermal tolerance, as evident from the dose response curves presented in Figure 2 and the newly added ED50 density distribution plot in the Supplement (ESM Figure 9). This is good news, as you infer, as it provides a prerequisite for thermal pre-screening (i.e., variation in thermal tolerance). We have added a table in the Supplement (ESM 8) to show the average reproducibility of colony-level ED50 values across timepoints (i.e., +/- difference in ED50, *sensu* Cunning et al., 2024 – <https://doi.org/10.1007/s00338-024-02577-7>).

Further to that, since large ED50 differences between colonies are needed to reproduce colony ranks over time accurately (see Cunning et al., 2024 follow up on the 2021 paper detailing this reproducibility) we sought to address it in a way to assess how many of the better-/worse-performing in situ colonies (top/bottom) retained their cohort assignment in the nursery for both time points and in consideration of the screening effort, i.e. number of assayed colonies – this, we think, may provide a pathway to application. We considered the top 5 and bottom 5 colonies (irrespective of site and coral species), amounting to the top/bottom 20% in our study and assessed consistency between in situ and nursery corals. As you can see from the figure below (new Figure 4 in the revised manuscript), even at a screening effort of 10 colonies, an accuracy of about 80% (<2 colonies overlap in 10 colonies) is achieved. The analysis also suggests that with a reasonable screening effort of 20 colonies, less than 1 colony is wrongly assigned to the top or bottom cohort. Interestingly, there is a larger discrepancy in the later time point which may be attributable to differences in dynamic recovery (not all corals ‘recover’ to their initial thermal tolerance at equal rates) and potential seasonal influences (see Garcia et al., 2024 -<https://www.nature.com/articles/s42003-024-07340-w>).

Figure 4. Assessment of assignment consistency into top- and bottom-performing colonies comparing in situ colonies to their nursery counterparts based on screening effort (number of assayed colonies). (a) October 2022 and (b) March 2023. Colonies were ranked by their ED50s, subsampled to different sample sizes, and it was

assessed whether the top and bottom five colonies were consistent between in situ and nursery corals. Mean overlap indicates the mean number of wrongly assigned colonies. SE = standard error.

Added text reads as follows:

Material and methods:

“Reproducibility of colony-specific ED50 values and rank/cohort assignments

To support the selection of individual coral colonies for restoration, the reproducibility of thermal tolerance thresholds (ED50s) by means of assignment into bottom and top performing colonies was investigated. The reproducibility of ED50 values on the colony level (i.e., for each genet, Cunnings et al., 2024) was tested by comparing October 2022 ED50s to March 2023 ED50 values for each colony, respectively. Additionally, we assessed consistency of cohort assignment (top- vs. bottom-performing colonies) as a function of screening effort (i.e., number of colonies). Colonies were ranked by their ED50s and the mean overlap of the top5 and bottom5 colonies comparing donor and nursery corals in October 2022 and March 2023 were determined using random subsampling to assess accuracy as a function of screening effort (number of colonies assayed).”

Results:

“ED50 reproducibility and rank/cohort assignment consistency

*Colony-level reproducibility of ED50 varied among species and conditions (i.e., Donor, Nursery). For *A. florida*, reproducibility was appreciably high as 57% of reef-based donor colonies (n=21) and 62% of nursery corals (n=21) re-tested at an ED50 difference $< \pm 1$ °C when considering ED50 across time points (i.e., October 2022 vs March 2023) (ESM 8). For *A. cytherea*, ED50 reproducibility was lower, as 50% of reef-based donor colonies (n=24) and 37% of nursery corals (n=19) re-tested at an ED50 difference $< \pm 1$ °C. This variation in ED50 reproducibility between species is further highlighted by the distribution of ED50 values for each site-condition group across time points (ESM 9). By comparison, cohort assignment into top vs. bottom performing colonies showed that ~1 colony was misassigned at a screening size of ≥ 15 colonies considering the top and bottom five colonies (Figure 4). ”*

Discussion L483-501:

*“Importantly, the reproducibility of colony-level ED50 is critical for restoration application to confidently select heat tolerant corals. Although the ED50 reproducibility here was lower compared to Cunnings et al., 2024 (i.e., 37-62% vs 90% of corals re-tested at ED50 difference of $< \pm 1$ °C), the observed rates are still appreciably high, specifically when considering the loss and recovery of *A. cytherea* nursery corals and known seasonal adjustments of ED50s of > 1 °C for *Acropora* species in the Red Sea (García et al., 2024). For real-world application, consistency of cohort assignment (higher vs. lower thermal tolerance) may ultimately become*

more important than high reproducibility of ED50s. Our results show that top and bottom cohort assignment is highly consistent between reef-based donor and nursery corals for both time points, with the October 2022 time point showing higher consistency than the March 2023 time point. This may be attributable to differences in thermal tolerance threshold recovery, seasonal acclimation, etc. At large, the results suggest that with a reasonable screening effort of 20 colonies and subsequent selection of the top performing colonies, high accuracy can be achieved, emphasizing the suitability of CBASS assays to differentiate between heat-tolerant and -susceptible colonies across reef sites and species. However, at this conjecture, it is vitally important to resolve how ED50 differences map onto ecological in situ bleaching outcomes. This requires investigations on coral thermal tolerance and resilience to prevailing stressors which must be conducted at a species and population level across large sample sizes and habitats.”

While you saw no significant correlation between growth rates and ED50s, *A. cytherea* did exhibit lower ED50s (at that time point) and higher growth per day than *A. florida*. Do you think that a tradeoff did exist between species?

Response: Correct, differences among species existed, suggesting differential life-history strategies. For example, *A. cytherea* is one of the fastest growing corals, so perhaps its more dynamic physiology and fast growth requires energy investment in processes that favour growth over heat stress resilience. Moreover, associations with heat tolerant algae were shown to cause growth trade-offs in some coral (<https://doi.org/10.1007/s00338-014-1216-4>), which may be a reason why growth of *A. florida* (more heat tolerant) is lower compared to *A. cytherea*. However, this is a wider discussion and topic we omitted discussing since the (modest) growth data presented was only recorded over 176 days, which we do not consider long enough to infer robust conclusions.

Figures:

Figure 3: I'm unclear on how ESM7 differs from Figure 3. If they show the same values and same statistical comparisons, then why do the HSD letters differ between the two?

Response: Figure 3 shows multiple comparison results for donor and nursery corals located *within the same site but across timepoints*. In contrast, ESM 7 compares nursery and donor colonies *across sites and within time points* (i.e., October and March are tested separately). This additional test presented in ESM7 was necessary to see if there was a (species-specific) difference in coral thermal tolerance due to site origin of the coral, since reef-scale environmental conditions may result in site-specific acclimatization, thus possibly influencing the thermal tolerance of nursery corals. The site-specific test results are not shown in Fig3, since the major focus is on comparing across timepoints within the same site to show retention vs recovery of coral thermal tolerance. To make it easier to compare these two figures, we further revised ESM7 by editing the colors of the bars in panel b and d to follow the same color coding as Fig3 and

additionally organized the order of the bars in the same fashion as in Fig3 (i.e., Donor, Nursery, Donor, Nursery).

ESM7: Could you include the species and season information stated in the legend in the figure as well to be more interpretable?

Response: Revised accordingly.

ESM8: Could you include R2 and p values on these plots?

Response: Revised accordingly. ESM8 is now ESM10 after adding the ED50 distribution density plot and ED50 reproducibility as mentioned above.

R

e Introduction

Reads well, to the point and well up-to-date with literature.

e M&M

Fig1 – can you at Lat Long coordinates to the map?

L119-121: The phrase "kept constant" is a bit unclear. I think you mean that the depth of the frame was kept at the same level as the average depth of the donor colonies. Maybe rephrase it to just articulate that a little clearer?

L126: How are length and height different? I understand you measured width, but the difference between length and height is unclear to me.

Line 148: Could you provide more ecological context for the chosen temperature treatments? Is 39.5°C a temperature these corals are actually exposed to during an extreme heatwave, or is it intended push the corals thus far to see a drop in their performance? It seems extremely high. Adding a brief explanation would help readers better interpret the relevance of the results.

Lines 149–155: I'm having some trouble following how the fragments were sampled. The "n=xxx" for each species is clear, but it would be helpful to know how many donor colonies this represents. For example, line 154 mentions four replicate fragments per donor colony—does this mean you sampled from three donor colonies (with four fragments each) to reach $n=12$ for *A. florida*? Clarifying this would make the sampling design easier to follow.

Lines 182–188: Am I right in understanding that you re-measured the same nursery fragments, but had to sample new fragments from the donor colonies at the sites? If not, how did you maintain the original fragments for re-measurement? A bit more explanation here would help clarify the methodology.

L206-208: I think you should start this paragraph with this definition of "coral thermal tolerance thresholds", rather than stating this halfway through this paragraph.

Statistical analyses look robust and good to me although I would not consider myself an expert in this domain.

Results

Fig2 – there is a little white square in panel a at x=40 degrees. Is this a formatting error?

How did you calculate the relative thermal threshold? I can't seem to find this in the methods. I read about how you got to ED50 but not the relative thermal threshold, which is needed to understand Fig3.

Discussion

L. 405–406 and 422–423: I'm unclear on the concept of pre-screening phenotypes through heat stress. As I understand it, your study did not conduct pre-screening—rather, the heat stress test was done after >150 days, and only then was a drop in Fv/Fm observed in one species. So what exactly is the pre-screening for? Is it to understand that although performance may dip temporarily it should return to pre-stocking/pre-screening levels? If so, this could be explained a little more clearly. Mostly, it would be helpful to better understand the purpose of pre-screening (I guess answer a little clearer the "so what" question).

Lines 16-17: Revise this sentence as is currently awkwardly worded e.g., "...to enhance restoration success is being considered by/for..."

Lines 20-21: I understand what's intended by "nursery offspring" here but I'm concerned it may be misinterpreted as actual offspring.

Line 45: And importantly marine heatwaves are becoming more frequent

Line 61: What is meant by "markers" to identify heat tolerant phenotypes? This suggests biomarkers or genetic markers which I wouldn't consider "globally advanced".

Lines 359-362: See Caruso et al 2025 ("Short-term stress testing predicts subsequent natural bleaching variation") for an update here, though your statement holds on how we are still missing a definitive understanding.

General results:

Was it unexpected that the 2 sites did not differ in their heat tolerance, given their environmental differences?

It looks like there was a wide variation in thermal tolerances within species/sites/conditions. As you study directly relates to choosing individuals for restoration, it would be interesting to see the distribution of these colony ED50s to understand if certain colonies/genotypes may be more affected than others by transplantation to the nursery. Also, was there a correlation in ED50s over time (as seen in Cunning et al "Census of heat tolerance among Florida's threatened staghorn corals finds resilient individuals throughout existing nursery populations" Figure 5)?

While you saw no significant correlation between growth rates and ED50s, *A. cytherea* did exhibit lower ED50s (at that time point) and higher growth per day than *A. florida*. Do you think that a tradeoff did exist between species?

Figures:

Figure 3: I'm unclear on how ESM7 differs from Figure 3. If they show the same values and same statistical comparisons, then why do the HSD letters differ between the two?

ESM7: Could you include the species and season information stated in the legend in the figure as well to be more interpretable?

ESM8: Could you include R2 and p values on these plots?